# Higher-Order Uncoupled Dynamics Do Not Lead to Nash Equilibrium — Except When They Do

**Sarah A. Toonsi**
Industrial and Enterprise Systems Engineering
University of Illinois Urbana-Champaign
Urbana, Illinois, USA
stoonsi2@illinois.edu

**Jeff S. Shamma**
Industrial and Enterprise Systems Engineering
University of Illinois Urbana-Champaign
Urbana, Illinois, USA
jshamma@illinois.edu

## Abstract

The framework of multi-agent learning explores the dynamics of how an agent's strategies evolve in response to the evolving strategies of other agents. Of particular interest is whether or not agent strategies converge to well-known solution concepts such as Nash Equilibrium (NE). In "higher-order" learning, agent dynamics include auxiliary states that can capture phenomena such as path dependencies. We introduce higher-order gradient play dynamics that resemble projected gradient ascent with auxiliary states. The dynamics are "payoff-based" and "uncoupled" in that each agent's dynamics depend on its own evolving payoff and has no explicit dependence on the utilities of other agents. We first show that for any polymatrix game with an isolated completely mixed-strategy NE, there exist higher-order gradient play dynamics that lead (locally) to that NE, both for the specific game and nearby games with perturbed utility functions. Conversely, we show that for any higher-order gradient play dynamics, there exists a game with a unique isolated completely mixed-strategy NE for which the dynamics do not lead to NE. Finally, we show that convergence to the mixed-strategy equilibrium in coordination games can come at the expense of the dynamics being inherently internally unstable.

## 1 Introduction

The field of learning in games explores how game-theoretic solution concepts emerge as the outcome of dynamic processes where agents adapt their strategies in response to the evolving strategies of other agents (1; 2; 3; 4). There is a multitude of specific cases of learning dynamics/game combinations that result in a range of outcomes, including convergence, limit cycles, chaotic behavior, and stochastic stability (5; 6; 7; 8; 9; 10; 11; 12; 13). The emphasis in the literature is on simple adaptive procedures, called "natural" dynamics in (14), that can result in various solution concepts (e.g., Nash equilibrium, correlated equilibrium, and coarse correlated equilibrium). (A separate concern is the complexity associated with such computations (e.g. (15; 16)).)

One "natural" restriction for learning is that the dynamics of one agent should not depend explicitly on the utility functions of other agents. This restriction was referred to as "uncoupled" dynamics in (17), where the authors constructed a specific anti-coordination matrix game for which no uncoupled learning dynamics could converge to the unique mixed-strategy NE. The dynamics considered in that setting were of fixed order, i.e., the order of the learning dynamics was restricted to match the dimension of the strategy space. More recent work showed that specific instances of fixed order uncoupled learning dynamics can never lead to mixed-strategy NE (18). Furthermore, there exist games for which any fixed order learning dynamics are bound to have an initial condition starting from which the dynamics do not converge to NE (19).

37th Conference on Neural Information Processing Systems (NeurIPS 2023).

The restriction on the order of the learning dynamics in learning mixed-strategy NE turns out to be essential. In particular, by introducing additional auxiliary states, (20) showed that higher-order learning could overcome the obstacle of convergence to NE in the same anti-coordination game considered in (17) while remaining uncoupled.

Higher-order learning in games can be seen as a parallel to higher-order optimization algorithms, such as momentum-based or optimistic gradient algorithms (e.g., (21; 22)). Such algorithms utilize history to update the underlying search parameter. In this way, there is a path dependency on the trajectory of information. An early utilization of higher-order learning is in (23), in which a player's strategy update uses two stages of history of an opponent's strategies in a zero-sum setting to eliminate oscillations. Similar ideas were used in (24). Reference (25) modified gradient-based algorithms through the introduction of a cumulative (integral) term. In (20), higher-order dynamics were used to create a myopic forecast of the action of other agents. Authors in (26) introduce a version of higher-order replicator dynamics and show that, unlike fixed order replicator dynamics, weakly dominated strategies become extinct. Reference (27) utilizes the system theoretic notion of passivity to analyze a family of higher-order dynamics.

In this paper, we further explore the implications of learning dynamics that are uncoupled. We address "payoff based" dynamics, in which the learning dynamics depend on the evolution of a payoff vector that is viewed as an externality. When players are engaged in a game, then the payoff stream of one agent depends on the actions of other agents. However, the learning dynamics themselves do not change based on the source of the payoff streams.

First, we show that for any polymatrix game with a mixed-strategy NE, there exist payoff-based dynamics that converge locally to that NE. This result is established by making a connection between convergence to NE and the existence of decentralized stabilizing control (28; 29). A consequence of the payoff-based structure is that the dynamics also converge to the NE of nearby perturbations of the original game. The form of higher-order learning used for this stability result is higher-order gradient play, which generalizes gradient ascent. Next, we show that for any such dynamics, there exists a game with a unique mixed-strategy NE that is unstable under given dynamics. The tool utilized is a classical analysis method in feedback control systems known as root-locus (e.g., (30), (31)), which characterizes the locations of the eigenvalues of a matrix as a function of a scalar parameter. A combination of the above results suggests the lack of universality on the side of both learning dynamics and games. While any mixed-strategy NE can be stabilized by suitable higher-order gradient dynamics, any such dynamics can be destabilized by a suitable anti-coordination game. Finally, we examine the implications of higher-order dynamics being able to converge to the mixed-strategy NE of a $2 \times 2$ coordination game, which has two pure NE and one mixed-strategy NE. We show that such higher-order gradient play dynamics must have an inherent internal instability, which makes them unsuitable, if not irrational, as a model of learning.

## 2 Payoff-based learning dynamics

### 2.1 Finite polymatrix games

We consider finite (normal form) games over mixed-strategies. There are $n$ players. The strategy space of player $i \in \{1, 2, ..., n\}$ is the probability simplex, $\Delta(k_i)$, where $k_i$ is a positive integer and $\Delta(\cdot)$ is defined as $\Delta(\kappa) = \left\{ s \in \mathbb{R}^\kappa \mid s_j \geq 0, j = 1, ..., \kappa, \ \& \ \sum_{j=1}^{\kappa} s_j = 1 \right\}$. The utility function of player $i$ is a function $u_i : \Delta(k_1) \times ... \times \Delta(k_n) \to \mathbb{R}$. We sometimes will write $u_i(x_1, ..., x_n) = u_i(x_i, x_{-i})$, for $x_i \in \Delta(k_i)$ and $x_{-i} \in \mathcal{X}_{-i}$, where $\mathcal{X}_{-i} = \Delta(k_1) \times ... \times \Delta(k_{i-1}) \times \Delta(k_{i+1}) \times ... \times \Delta(k_n)$.

For convenience, we will restrict our discussion to pairwise interactions, also called polymatrix games. That is, the utility function of player $i$ is defined as

$$u_i(x_1, ..., x_n) = x_i^{\mathrm{T}} \sum_{\substack{j=1 \\ j \neq i}}^{n} M_{ij} x_j, \tag{1}$$

for matrices, $M_{ij}, j = 1, \ldots, n, j \neq i$. We can write the utility function of player $i$ as the inner product $u_i(x_1, ..., x_n) = x_i^{\mathrm{T}} P_i(x_{-i})$ where $P_i(x_{-i}) = \sum_{\substack{j=1 \\ j \neq i}}^{n} M_{ij} x_j \in \mathbb{R}^{k_i}$. Accordingly, each element of $P_i(x_{-i})$ can be viewed as a payoff that is associated with a component of player $i$'s strategy vector, $x_i$.

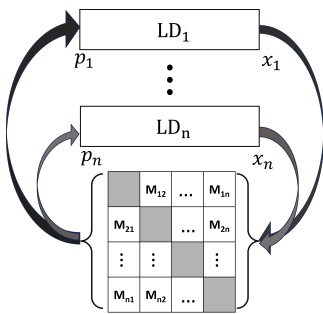

Figure 1: Payoff-based learning dynamics ($\text{LD}_i$) in feedback with game matrices ($M_{ij}$).

A Nash equilibrium (NE) is a tuple $(x_1^*, ..., x_n^*) \in \mathcal{X}$ such that for all $i = 1, ..., n$,

$$u_i(x_i^*, x_{-i}^*) \geq u_i(x_i, x_{-i}^*), \quad \forall x_i \in \Delta(k_i).$$

A completely mixed-strategy NE is such that each $x_i^*$ is in the interior of the simplex.

## 2.2 Fixed order learning

Our model of learning is a dynamical system that relates trajectories of a payoff vector, $p_i(t)$, to trajectories of the strategy, $x_i(t)$. In particular, learning dynamics for player $i$ are specified by a function $f_i : \Delta(k_i) \times \mathbb{R}^{k_i} \to \mathbb{R}^{k_i}$ according to

$$\dot{x}_i(t) = f_i(x_i(t), p_i(t)),$$

where $x_i(t) \in \Delta(k_i)$ and $p_i : \mathbb{R}_+ \to \mathbb{R}^{k_i}$. We assume implicitly that $f_i$ and $p_i$ are such that there exists a unique solution whenever $x_i(0) \in \Delta(k_i)$. We further assume that the dynamics satisfy the invariance property that

$$x_i(0) \in \Delta(k_i) \Rightarrow x_i(t) \in \Delta(k_i), \quad \forall t \geq 0. \tag{2}$$

Note that we define learning dynamics without specifying the source of the payoff vector, $p_i(t)$, hence the terminology "payoff-based". Only once a player is coupled with other players in a game through their own (possibly heterogeneous) learning dynamics is when we make the connection $p_i(t) = P_i(x_{-i}(t))$.

This formulation is illustrated in Figure 1, where the $\text{LD}_i$ denote payoff-based learning dynamics that are interconnected through the game matrices, $M_{ij}$. Note that such learning dynamics are uncoupled by construction since each player can only access its own payoff vector. There is no dependence on the payoff stream of other players. Indeed, there is no dependence on the parameters of one's own utility function.

## 2.3 Higher-order learning

The learning dynamics described in the previous section have a fixed order associated with the dimension of the strategy space. Higher-order learning dynamics allow for the introduction of auxiliary states as follows. For any fixed order learning dynamics, $f_i$, we can define a higher-order version as

$$\dot{x}_i(t) = f_i(x_i(t), p_i(t) + \phi_i(p_i(t), z_i(t))) \tag{3a}$$
$$\dot{z}_i(t) = g_i(p_i(t), z_i(t)). \tag{3b}$$

As before, $x_i(t) \in \Delta(k_i)$ and $p_i : \mathbb{R}_+ \to \mathbb{R}^{k_i}$. The new variable $z_i \in \mathbb{R}^{\ell_i}$ represents $\ell_i$ dimensional auxiliary states that evolve according to the $p_i$-dependent dynamics, $g_i$. These enter into the original fixed order dynamics through $\phi_i$. Accordingly, we can view $p_i(t) + \phi_i(p_i(t), z_i(t))$ as a modified payoff stream that captures path dependencies in $p_i$, and the original learning dynamics react to this modified payoff stream.

As with the fixed order counterparts, there is no specification of game parameters in higher-order learning dynamics. In order to enforce that the auxiliary states have no effect on the equilibria of games, we make the following assumption.

**Assumption 2.1.** *If $p_i^*$ and $z_i^*$ are an equilibrium of the higher-order dynamics, i.e.,*

$$0 = g_i(p_i^*, z_i^*)$$

*then*

$$\phi_i(p_i^*, z_i^*) = 0.$$

This assumption assures that the auxiliary states represent purely transient phenomena that disappear at equilibrium.

**Example 1** (**Anticipatory higher-order dynamics**). *A special case of higher-order dynamics is*

$$\dot{z}_i = \lambda(p_i - z_i)$$
$$\phi_i = \gamma\lambda(p_i - z_i),$$

*where the the higher-order modification is the linear system (see Appendix A.1)*

$$\left(\begin{array}{c|c} -\lambda I & \lambda I \\ \hline -\gamma\lambda I & \gamma\lambda I \end{array}\right),$$

*and $\gamma, \lambda \in \mathbb{R}_+$. Intuition behind the connection to anticipation can be seen by viewing $\lambda(p_i - z_i)$ as an approximation of $\dot{p}_i$, and so $p_i(t) + \gamma\lambda(p_i(t) - z_i(t)) \approx p_i(t + \gamma)$. Similar higher-order dynamics were used in reference (20) for smooth fictitious play to overcome the lack of convergence of uncoupled dynamics to NE in the anti-coordination game analyzed by (17). See also reference (32) for an analysis with replicator dynamics.*

*Anticipatory higher-order dynamics can also be linked to optimistic optimization algorithms (e.g., (22)). An Euler discretization of step size, $h$, results in*

$$z_i^+ = z_i + h\lambda(p_i - z_i),$$

*with a modified payoff stream of*

$$p_i + \phi_i = p_i + \gamma\lambda(p_i - z_i).$$

*Setting $h = 1/\lambda$ and $\gamma = 1/\lambda$ results in*

$$p_i + \phi_i = p_i + (p_i - z_i)$$
$$= p_i + (p_i - p_i^-),$$

*where the superscripts '+' and '−' indicate the next and previous discrete time steps, respectively. There are also optimistic variants of discrete-time no-regret learning algorithms (33; 34; 35) that guarantee faster convergence rates to coarse correlated equilibria compared to the standard versions.*

## 3 Gradient play

The main results of the paper will examine the behavior of gradient play and higher-order gradient play, which are the focus of this section.

### 3.1 Fixed order gradient play

In gradient play dynamics, a player adjusts its strategy in the direction of the payoff stream, i.e.,

$$\dot{x}_i = \Pi_\Delta[x_i + p_i] - x_i, \tag{4}$$

where $\Pi_\Delta[x] : \mathbb{R}^n \to \Delta(n)$ is the projection of $x$ into the simplex, i.e., $\Pi_\Delta(x) = \arg\min_{s \in \Delta(n)} \|x - s\|$.

The terminology "gradient play" stems from the gradient of an agent's utility function in (1) with respect to its own strategy, $x_i$, namely $\nabla_{x_i} u_i(x_i, x_{-i}) = P_i(x_{-i}) = \sum_{\substack{j=1 \\ j \neq i}}^n M_{ij}x_j$. As was done in the description of payoff-based learning, we replace $P_i(x_{-i})$ with the payoff stream $p_i$ without regard to the game matrices $M_{ij}$.

Our primary concern will be studying these dynamics near a completely mixed-strategy NE. To this end, let $x^* = (x_1^*, \ldots, x_n^*)$ be an isolated completely mixed-strategy NE. The strategy vector $x_i$

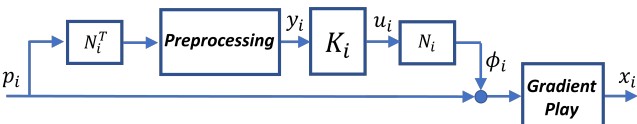

Figure 2: Cascade representation of linear higher-order dynamics for gradient play.

evolves on the simplex, which is a subset of dimension $k_i - 1$. Hence, the local behavior of the dynamics around $x_i^*$ is characterized by evolution on a lower-dimensional subset. Thus, we can write

$$x_i = x_i^* + N_i w_i \tag{5}$$

where

$$\mathbf{1}^\mathrm{T} N_i = 0 \ \& \ N_i^\mathrm{T} N_i = I. \tag{6}$$

Therefore, $w_i \in \mathbb{R}^{(k_i-1)}$ represents deviations from $x_i^*$ and satisfies $w_i(t) = N_i^\mathrm{T}(x_i(t) - x_i^*)$. When all players utilize fixed order gradient play, the collective dynamics near a completely mixed-strategy NE take the form

$$\dot{w} = \mathcal{M}w, \tag{7}$$

where

$$\mathcal{M} = \mathcal{N}^\mathrm{T} \begin{pmatrix} 0 & M_{12} & M_{13} & \dots & M_{1n} \\ M_{21} & 0 & M_{23} & \dots & M_{2n} \\ M_{31} & M_{32} & 0 & \dots & M_{3n} \\ \vdots & \vdots & \vdots & \ddots & \vdots \\ M_{n1} & M_{n2} & M_{n3} & \dots & 0 \end{pmatrix} \mathcal{N}, \quad \mathcal{N} = \begin{pmatrix} N_1 & & \\ & \ddots & \\ & & N_n \end{pmatrix}. \tag{8}$$

Given the zero trace of $\mathcal{M}$, standard gradient play is always unstable at a completely mixed-strategy NE (see Appendix A.3 for stability conditions). Also, for this equilibrium to be isolated, $\mathcal{M}$ must be non-singular. Otherwise, $\mathcal{M}$ has a non-trivial null space leading to an equilibrium subspace.

## 3.2 Higher-order gradient play

We will be interested in a specific form of higher-order gradient play that uses the following linear structure of higher-order dynamics:

$$\dot{x}_i = -x_i + \Pi_\Delta \left[ x_i + p_i + N_i(G_i \xi_i + H_i(N_i^\mathrm{T} p_i - v_i)) \right]$$
$$\dot{\xi}_i = E_i \xi_i + F_i(N_i^\mathrm{T} p_i - v_i)$$
$$\dot{v}_i = N_i^\mathrm{T} p_i - v_i,$$

for some matrices $E_i$, $F_i$, $G_i$ and $H_i$. Here, the auxiliary states are $z_i = (v_i, \xi_i)$, which enter into the dynamics through $\phi_i(p_i, \xi_i, v_i) = N_i(G_i \xi_i + H_i(N_i^\mathrm{T} p_i - v_i))$, where $N_i$ is defined as in (6).

The motivation behind this structure, illustrated in Figure 2, assures the enforcement of Assumption 2.1. The payoff stream is first preprocessed by a specific linear system to produce $y_i$ and then by a general linear system $K_i$ to produce $u_i$. The matrices $(E_i, F_i, G_i, H_i)$ create the dynamical system

$$K_i \sim \left( \begin{array}{c|c} E_i & F_i \\ \hline G_i & H_i \end{array} \right) \tag{9}$$

that maps $y_i = N_i^\mathrm{T} p_i - v_i$ to $u_i = G_i \xi_i + H_i y_i$ via $\dot{\xi}_i = E_i \xi_i + F_i y_i$. The preprocessing system has the property (see "washout filters" in Appendix A.2) that if $p_i(t)$ converges to a constant, then $y_i(t)$ converges to zero. Accordingly, when $E_i$ is non-singular, the preprocessing system guarantees Nash stationarity, i.e., if

$$\lim_{t \to \infty} (x_i(t), \xi_i(t), v_i(t)) = (x_i^*, \xi_i^*, v_i^*) \quad \forall i$$

then $x^* = (x_1^*, \dots, x_n^*)$ is a NE. To conclude, the linear system $K_i$ is the central entity that performs the payoff modification, and the preprocessing system ensures compliance with Assumption 2.1.

## 3.3 Local stability analysis

As before, we can analyze the behavior near a completely mixed-strategy NE $x^*$ through the variable $w_i$ defined as in (5). Using the fact that $N_i^{\mathrm{T}} \sum_{j \neq i} M_{ij} x_j^* = 0$, we can write the local dynamics of a player as

$$\dot{w}_i = N_i^{\mathrm{T}} \sum_{j \neq i} M_{ij} N_j w_j + G_i \xi_i + H_i \left( N_i^{\mathrm{T}} \sum_{j \neq i} M_{ij} N_j w_j - v_i \right)$$

$$\dot{\xi}_i = E_i \xi_i + F_i \left( N_i^{\mathrm{T}} \sum_{j \neq i} M_{ij} N_j w_j - v_i \right)$$

$$\dot{v}_i = N_i^{\mathrm{T}} \sum_{j \neq i} M_{ij} N_j w_j - v_i.$$

The collective dynamics near a mixed-strategy NE can be written as

$$\begin{pmatrix} \dot{w} \\ \dot{\xi} \\ \dot{v} \end{pmatrix} = \begin{pmatrix} (I+H)\mathcal{M} & G & -H \\ F\mathcal{M} & E & -F \\ \mathcal{M} & 0 & -I \end{pmatrix} \begin{pmatrix} w \\ \xi \\ v \end{pmatrix}, \tag{10}$$

where $E$, $F$, $G$, and $H$ are block diagonal matrcies with appropriate dimensions and $\mathcal{M}$ is defined in (8). Local stability of a completely mixed NE is determined by whether the above collective dynamics are stable, i.e., the dynamics matrix in (10) is a stability matrix.

# 4 Uncoupled dynamics that lead to mixed-strategy NE

## 4.1 Decentralized control formulation

The stability of a mixed-strategy equilibrium is tied to the existence of $K_1$, $K_2$, ..., $K_n$ so that the linear system in (10) is stable. When the $K_i$ have yet to be determined, we can rewrite (10) as

$$\begin{pmatrix} \dot{w} \\ \dot{v} \end{pmatrix} = \begin{pmatrix} \mathcal{M} & 0 \\ \mathcal{M} & -I \end{pmatrix} \begin{pmatrix} w \\ v \end{pmatrix} + \begin{pmatrix} I \\ 0 \end{pmatrix} u, \tag{11a}$$

$$y = \begin{pmatrix} \mathcal{M} & -I \end{pmatrix} \begin{pmatrix} w \\ v \end{pmatrix}, \tag{11b}$$

where $u = \begin{pmatrix} u_1 \\ \vdots \\ u_n \end{pmatrix}$, $y = \begin{pmatrix} y_1 \\ \vdots \\ y_n \end{pmatrix}$, and the $y_i$ and $u_i$ are to be related through $K_i$.

## 4.2 Decentralized stabilization

Let

$$\mathcal{P} \sim \left( \begin{array}{c|c} \mathcal{A} & \mathcal{B} \\ \hline \mathcal{C} & 0 \end{array} \right)$$

with

$$\mathcal{A} = \begin{pmatrix} \mathcal{M} & 0 \\ \mathcal{M} & -I \end{pmatrix}, \quad \mathcal{B} = \begin{pmatrix} I \\ 0 \end{pmatrix}, \quad \mathcal{C} = \begin{pmatrix} \mathcal{M} & -I \end{pmatrix}. \tag{12}$$

We first establish that $\mathcal{P}$ can be stabilized by verifying the conditions for stabilizability and detectability (see Appendix A.3). The assumption that $\mathcal{M}$ is non-singular stems from our interest in isolated NE.

**Proposition 4.1.** *For $\mathcal{M}$ non-singular, the pair $(\mathcal{A}, \mathcal{B})$ is stabilizable, and the pair $(\mathcal{A}, \mathcal{C})$ is detectable.*

While Proposition 4.1 establishes that $\mathcal{P}$ can be stabilized, that property alone is inadequate for our purposes. In particular, for the learning dynamics to be uncoupled, we seek to establish decentralized stabilization (see Appendix A.4) according to the partition

$$\begin{pmatrix} \dot{w} \\ \dot{v} \end{pmatrix} = \mathcal{A} \begin{pmatrix} w \\ v \end{pmatrix} + \sum_{i=1}^{n} \mathcal{B}_i u_i, \quad y_i = \mathcal{C}_i \begin{pmatrix} w \\ v \end{pmatrix}, \tag{13}$$

where

$$\mathcal{A} = \begin{pmatrix} \mathcal{M} & 0 \\ \mathcal{M} & -I \end{pmatrix}, \quad \mathcal{B}_i = \begin{pmatrix} \mathcal{E}_i \\ 0 \end{pmatrix}, \quad \mathcal{C}_i = \begin{pmatrix} \mathcal{M}_{i\bullet} & -\mathcal{E}_i^{\mathrm{T}} \end{pmatrix}. \tag{14}$$

Here, $\mathcal{M}_{i\bullet}$ denotes the $i^{\text{th}}$ block row of $\mathcal{M}$, i.e.,

$$\mathcal{M}_{i\bullet} = \begin{pmatrix} N_i^{\mathrm{T}} M_{i1} N_1 & \dots & N_i^{\mathrm{T}} M_{i(i-1)} N_{i-1} & 0 & N_i^{\mathrm{T}} M_{i(i+1)} N_{i+1} & \dots & N_i^{\mathrm{T}} M_{in} N_n \end{pmatrix}$$

and

$$\mathcal{E}_i^{\mathrm{T}} = \begin{pmatrix} 0 & \dots & 0 & \underbrace{I}_{i^{\text{th}} \text{ position}} & 0 & \dots & 0 \end{pmatrix},$$

where $I$ has a dimension (suppressed in the notation) of $k_i - 1$.

**Theorem 4.1.** *For any isolated (i.e., $\mathcal{M}$ is non-singular) completely mixed-strategy NE, there exist uncoupled higher-order gradient play dynamics such that (10) is stable.*

The proof of Theorem 4.1 relies on the conditions of Theorem A.1 and is presented in Appendix B.1.

Theorem 4.1 should be viewed as a statement regarding whether uncoupled learning in itself is a barrier to learning dynamics leading to NE. The theorem makes no claim that the higher-order learning dynamics are interpretable (e.g., as in anticipatory learning). Nor does the theorem offer guidance on how agents may construct the matrices of higher-order learning that lead to convergence. In the next section, we will see that, while the structure is universal, any specific set of parameters is not universal in that one can construct a game for which they do not lead to NE. Despite the lack of universality, there is an inherent robustness that is a consequence of stability. The following follows from standard arguments on linear systems.

**Proposition 4.2.** *Let the $K_i$ and $M_{ij}$, $i = 1, ..., n$ and $j = 1, ..., n$, be such that (10) is stable. Then there exists a $\delta > 0$ such that (10) is stable with the $M_{ij}$ replaced by any $\tilde{M}_{ij}$ as long as $\left\| \tilde{M}_{ij} - M_{ij} \right\| < \delta$ for all $i = 1, ..., n$ and $j = 1, ..., n$.*

In words, this proposition guarantees that learning dynamics that lead to NE for a specific game continue to do so for nearby games.

### 4.3 Stabilization through a single higher-order player

The previous section's analysis allowed all players to utilize higher-order learning. In some cases, it may not be necessary for all players to utilize higher-order learning. In this section, we present sufficient conditions under which a single player using higher-order gradient play with the remainder utilizing fixed order gradient play can still lead to NE.

**Assumption 4.1.**

A. *Let $(w, \lambda)$ be a left eigenvalue pair of $\mathcal{M}$, i.e.,*

$$w^{\mathrm{T}} \mathcal{M} = \lambda w^{\mathrm{T}},$$

*with $\mathbf{Re}[\lambda] \geq 0$ and $w^{\mathrm{T}} = \begin{pmatrix} w_1^{\mathrm{T}} & \dots & w_n^{\mathrm{T}} \end{pmatrix}$ partitioned consistently with (8). Then $w_i \neq 0$ for all $i$.*

B. *Let $(v, \lambda)$ be a right eigenvalue pair of $\mathcal{M}$, i.e.,*

$$\mathcal{M} v = \lambda v,$$

*with $\mathbf{Re}[\lambda] \geq 0$ and $v = \begin{pmatrix} v_1^{\mathrm{T}} & \dots & v_n^{\mathrm{T}} \end{pmatrix}^{\mathrm{T}}$ partitioned consistently with (8). Then $v_i \neq 0$ for all $i$.*

Recall the definitions of $\mathcal{A}$, $\mathcal{B}_i$, and $\mathcal{C}_i$ from (14).

**Proposition 4.3.** *Let $\mathcal{M}$ be non-singular and satisfy Assumption 4.1. Then for any $i$, the pair $(\mathcal{A}, \mathcal{B}_i)$ is stabilizable and the pair $(\mathcal{A}, \mathcal{C}_i)$ is detectable.*

As a consequence of Proposition 4.3, it is possible for a completely mixed-strategy NE to be stabilized where a single player utilizes higher-order gradient play with the remaining players utilizing fixed order gradient play.

## 5 Non-convergence to NE in higher-order gradient play

We now show that linear higher-order gradient play dynamics need not lead to NE. Given any such dynamics, we construct a game with a unique NE that is unstable under given dynamics.

### 5.1 The Jordan anti-coordination game

The Jordan anti-coordination game, introduced in (36), was used in (17) to prove that fixed order uncoupled learning dynamics do not lead to NE. The game consists of three players with

$$u_1(x_1, x_2) = x_1^{\mathrm{T}} \begin{pmatrix} 0 & 1 \\ 1 & 0 \end{pmatrix} x_2, \quad u_2(x_2, x_3) = x_2^{\mathrm{T}} \begin{pmatrix} 0 & 1 \\ 1 & 0 \end{pmatrix} x_3, \quad u_3(x_3, x_1) = x_3^{\mathrm{T}} \begin{pmatrix} 0 & 1 \\ 1 & 0 \end{pmatrix} x_1,$$

and a unique mixed-strategy NE at $x_1^* = x_2^* = x_3^* = \left( \frac{1}{2} \quad \frac{1}{2} \right)^{\mathrm{T}}$. We will let $\Gamma(\mu)$ denote the Jordan anti-coordination game but with the utility function of player 1 modified to $u_1(x_1, x_2) = x_1^{\mathrm{T}}(\mu M_{12})x_2$, where $\mu \in \mathbb{R}_+$. Since scaling payoffs does not change the nature of the game, $\Gamma(\mu)$ has the same unique NE as $\Gamma(1)$.

### 5.2 Destabilization using rescaled anti-coordination

Suppose all three players use variants of linear higher-order gradient play in $\Gamma(\mu)$. As before, we denote the higher-order dynamics of player $i$ as the linear dynamical system (9). To study the local behavior of the dynamics around the unique mixed-strategy NE of $\Gamma(\mu)$, we define $w_1(t)$, $w_2(t)$, and $w_3(t)$ as in (5). Then, we analyze the local stability of the mixed-strategy NE through (10).

The following lemma will be essential in proving our main result.

**Lemma 5.1.** *Let $A \in \mathbb{R}^{n \times n}$, $B \in \mathbb{R}^{n \times 1}$ and $C \in \mathbb{R}^{1 \times n}$. If $CB = CAB = 0$, and $CA^m B \neq 0$ for some $m \geq 2$. Then for sufficiently large $\mu > 0$, $A - \mu BC$ is not a stability matrix.*

The proof of Lemma 5.1 is presented in Appendix B.2 and it uses root-locus arguments (see references (31; 30)).

**Proposition 5.1.** *If linear higher-order gradient play dynamics are locally exponentially stable at the unique NE of $\Gamma(1)$, then there exists $\mu > 0$ such that the unique NE of $\Gamma(\mu)$ is unstable under such dynamics.*

The proof of Proposition 5.1 is presented in Appendix B.3. The main idea of the proof is to write the local dynamics matrix in the form $A - \mu BC$, and then use Lemma 5.1. We also provide a similar proof for sufficiently small $\mu$ in Appendix C.

The results might be puzzling because, for all $\mu > 0$, all games $\Gamma(\mu)$ are strategically equivalent. Convergence guarantees for learning dynamics are usually established amongst classes of games. Thus, it is generally expected that dynamics will behave similarly for all games in a particular class. In this case, we design linear learning dynamics that are affected by simple rescaling of the payoff matrices.

## 6 Strong stabilization of mixed-strategy NE

Results from Section 4 imply that the mixed-strategy NE in a two-player $2 \times 2$ (identical-interest) coordination game can be stabilized. Here, we argue why dynamics that stabilize this mixed-strategy equilibrium are not reasonable. Specifically, we show that such dynamics *must be* inherently unstable as an open system, i.e., as dynamics that respond to an exogenous payoff stream, and this instability is problematic with respect to such payoffs.

First, we inspect which type of mixed-strategy NE requires unstable learning dynamics for stabilization. For this purpose, consider the system in (12) for $n = k_1 = k_2 = 2$:

$$\mathcal{A} = \begin{pmatrix} 0 & m_{12} & 0 & 0 \\ m_{21} & 0 & 0 & 0 \\ 0 & m_{12} & -1 & 0 \\ m_{21} & 0 & 0 & -1 \end{pmatrix} \quad \mathcal{B} = \begin{pmatrix} 1 & 0 \\ 0 & 1 \\ 0 & 0 \\ 0 & 0 \end{pmatrix} \quad \mathcal{C} = \begin{pmatrix} 0 & m_{12} & -1 & 0 \\ m_{21} & 0 & 0 & -1 \end{pmatrix}. \tag{15}$$

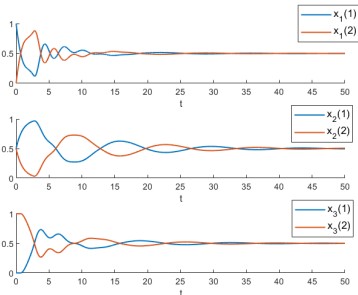

Figure 3: Single-player stabilization of the Jordan game.

Around an isolated mixed-strategy NE, the matrix $\mathcal{A}$ above should be non-singular. Accordingly, it must be that $m_{12} \neq 0$ and $m_{21} \neq 0$. The ability to stabilize a system via another stable system is referred to as *strong stabilization* (see Appendix A.3). The next proposition gives a sufficient condition under which an isolated mixed-strategy NE is not strongly stabilizable.

**Proposition 6.1.** *If $m_{12}m_{21} > 0$, then* (15) *is not strongly stabilizable.*

The proof of Proposition 6.1 uses the "parity interlacing principle" (see reference (37)) and is presented in Appendix B.4.

The nature of the game can be inferred from the scalars $m_{12}$ and $m_{21}$. For example in zero-sum games we have $M_{12} = -M_{21}^{\mathrm{T}}$, which gives

$$m_{12} = N^{\mathrm{T}} M_{12} N = N^{\mathrm{T}} M_{12}^{\mathrm{T}} N = -N^{\mathrm{T}} M_{21} N = -m_{21}.$$

In coordination games, we have $M_{12} = M_{21}$, which gives

$$m_{12} = N^{\mathrm{T}} M_{12} N = N^{\mathrm{T}} M_{21} N = m_{21}.$$

Therefore, the mixed-strategy NE in a coordination game is not strongly stabilizable.

Now let us examine the implications of inherently unstable learning dynamics. A reasonable expectation of learning dynamics is that in the case of a constant payoff vector, i.e., $p_i(t) \equiv p^*$, we expect

$$\lim_{t \to \infty} x_i(t) = \beta(p^*),$$

where $\beta(p^*)$ is a best response to $p^*$, i.e.,

$$\beta(p^*) = \arg\max_{x_i \in \Delta(k_i)} x_i^{\mathrm{T}} p^*.$$

For any higher-order gradient play dynamics, if $E_i$ is a stability matrix, then whenever $p_i(t) \equiv p^*$ for some constant vector $p^*$, one can show $\xi_i(t) \to 0$, which implies that $x_i(t)$ is generated by standard gradient play dynamics in the limit. However, if $p_i(t) \equiv p^*$ and the dynamics are inherently unstable, the term $N_i G_i \xi_i(t)$ need not vanish. Indeed, one can construct $p^*$ such that $x_i(t)$ does not converge to the best response of $p^*$ (see the example in Section 7.2). The inability of learning dynamics to converge to the best response of a constant payoff vector does not reflect "natural" behavior.

# 7 Numerical experiments

## 7.1 Jordan anti-coordination game: Stabilization through a single player

The payoff matrices of the Jordan anti-coordination game, introduced in Section 5.1, satisfy Assumption 4.1. Therefore, we can stabilize its mixed-strategy NE, allowing only one player to use higher-order learning while others continue to use standard gradient play. For this purpose, let $\xi_1 \in \mathbb{R}$ and choose $H_1 = \gamma\lambda$, $G_1 = -\gamma\lambda$, $F_1 = \lambda$ and $E_1 = -\lambda$, where $\lambda = 50$ and $\gamma = 5$. Such dynamics resemble anticipatory gradient play but on the filtered low-dimensional payoff. Figure 3 illustrates convergence of the players' strategies to NE.

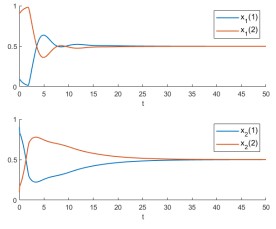

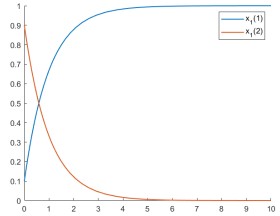

(a) Stabilization of the mixed-strategy NE.

(b) Inherently unstable dynamics do not converge to best response of $p^* = \begin{pmatrix} 0 & 1 \end{pmatrix}^{\mathrm{T}}$.

Figure 4: Stabilizing the mixed-strategy equilibrium of a coordination game and its consequences.

## 7.2 Stabilization of mixed-strategy NE in coordination games

Consider the (identical interest) coordination game:

$$u_1(x_1, x_2) = x_1^{\mathrm{T}} \begin{pmatrix} 1 & 0 \\ 0 & 1 \end{pmatrix} x_2, \quad u_2(x_2, x_1) = x_2^{\mathrm{T}} \begin{pmatrix} 1 & 0 \\ 0 & 1 \end{pmatrix} x_1.$$

The game has two pure strategy NE and one completely mixed-strategy NE at $x^* = \begin{pmatrix} \frac{1}{2} & \frac{1}{2} \end{pmatrix}^{\mathrm{T}}$. Consider the following set of parameters for higher-order gradient play: $E_1 = \lambda$, $F_1 = -2\lambda$, $G_1 = \gamma\lambda$, $H_1 = -\gamma\lambda$, $E_2 = -\lambda_2$, $F_2 = \lambda_2$, $G_2 = -\gamma_2\lambda_2$, and $H_2 = \gamma_2\lambda_2$. The numerical values are $\lambda = 0.5$, $\gamma = 20$, $\lambda_2 = 50$, and $\gamma_2 = 1$. Figure 4a illustrates convergence to the mixed-strategy NE of this coordination game. Suppose we break the feedback loop and use $p^* = \begin{pmatrix} 0 & 1 \end{pmatrix}^{\mathrm{T}}$ as the input to the first player dynamics. The response to such input is illustrated in Figure 4b. We see that $E_1 > 0$ is a scalar, and so $\xi_1$ grows without bound. The strategy $x_1$, which is projected to the simplex, converges to $\begin{pmatrix} 1 & 0 \end{pmatrix}^{\mathrm{T}}$, which is not a best response to the input $p^*$.

## 8 Concluding remarks

To recap, we studied the role of higher-order gradient play with linear higher-order dynamics. We showed that for any game with an isolated completely mixed-strategy NE, there exist higher-order gradient play dynamics that lead to that NE, both for the original game and for nearby games. On the other hand, we showed that for any higher-order gradient play dynamics, the dynamics do not lead to NE for a suitably rescaled anti-coordination game. We also provided an argument against dynamics that lead to the mixed-strategy NE in a coordination game, showing they are not reasonable.

Regarding the higher-order gradient play dynamics that lead to NE, the interpretation of the results herein should not be that these dynamics are either a descriptive model of learning or a prescriptive recommendation for computation. Rather, the results are a contribution towards delineating what is possible or impossible in multi-agent learning. In that sense, they may be seen as a complement to the contributions in (17). Namely, dynamics being uncoupled is not a barrier to converging to mixed-strategy NE when allowing higher-order learning.

More generally, the present results open new questions related to the discussion in (14) on what constitutes "natural" learning dynamics. In the case of anticipatory higher-order learning, there is a clear interpretation of the effect of higher-order terms. However, it is unclear how to interpret general higher-order dynamics. Furthermore, the results herein regarding inherent instability of higher-order dynamics that converge to the mixed-strategy NE of a coordination game suggests that higher-order learning can be "unnatural". Possible restrictions on dynamics, in addition to being uncoupled, could include having no asymptotic regret; maintaining qualitative behavior in the face of strategically equivalent games (cf., Section 5.2); or having an interpretable relationship between payoff streams and strategic evolutions such as "passivity", which generalizes and extends the notion of contractive games to contractive learning dynamics (e.g., (38; 39; 40)).

In terms of limitations, the current paper only addresses the payoff vector setup and does not address the setup where players only have access to instantaneous scalar payoffs. Furthermore, results in Sections 5 and 6 are limited to certain setups and require further generalization.

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

## Appendices

In Appendix A, we review some relevant topics from linear systems theory. In Appendix B, we provide proofs of theorems, lemmas, and propositions that were presented in the paper. In Appendix C, we provide an alternative destabilization proof for sufficiently small gain $\mu$.

## A  Background on linear systems

Here, we review some standard background material for linear dynamics systems. There are several references (e.g., (41; 42)) with more detailed exposition.

### A.1  Notation

The partitioned matrix

$$\left( \begin{array}{c|c} A & B \\ \hline C & D \end{array} \right)$$

represents a general linear dynamical system

$$\dot{x}(t) = Ax(t) + Bu(t), \quad x(0) = x_o \tag{16a}$$
$$y(t) = Cx(t) + Du(t) \tag{16b}$$

whose solution is

$$y(t) = Ce^{At}x_o + \int_0^t Ce^{A(t-\tau)}Bu(\tau)\,d\tau + Du(t).$$

The variables $x \in \mathbb{R}^n$, $u \in \mathbb{R}^m$, and $y \in \mathbb{R}^p$ here are used temporarily as generic placeholders (and will play a different role in the ensuing discussion). The notation $P \sim \left( \begin{array}{c|c} A & B \\ \hline C & D \end{array} \right)$ assigns the label $P$ to the dynamics (16).

### A.2  Washout filters

In the special case of

$$P \sim \left( \begin{array}{c|c} -I & I \\ \hline -I & I \end{array} \right)$$

where

$$\dot{x}(t) = -x(t) + u(t), \quad x(0) = x_o$$
$$y(t) = -x(t) + u(t)$$

and $x, u, y \in \mathbb{R}^n$, it is straightforward to show that if

$$\lim_{t\to\infty} u(t) = u^*,$$

then

$$\lim_{t\to\infty} y(t) = 0.$$

Linear systems with this property are known as "washout filters". For an extended discussion, see reference (43).

### A.3  Stability and stabilization

A matrix, $M$, is a *stability matrix* if, for every eigenvalue, $\lambda$, of $M$, $\mathbf{Re}[\lambda] < 0$.

The linear system, $P \sim \left( \begin{array}{c|c} A & B \\ \hline C & D \end{array} \right)$ is *stable* if $A$ is a stability matrix.

The linear system $K \sim \left( \begin{array}{c|c} E & F \\ \hline G & H \end{array} \right)$ *stabilizes* $P$ if the combined linear dynamics

$$\dot{x} = Ax + Bu$$
$$\dot{\xi} = E\xi + Fy$$
$$u = G\xi + Hy$$
$$y = Cx$$

or

$$\begin{pmatrix} \dot{x} \\ \dot{\xi} \end{pmatrix} = \underbrace{\begin{pmatrix} A + BHC & BG \\ FC & E \end{pmatrix}}_{M} \begin{pmatrix} x \\ \xi \end{pmatrix}$$

are stable, i.e., $M$ is a stability matrix.

The following conditions are necessary and sufficient for the existence of such a $K$. For all complex $\lambda$ with $\mathbf{Re}[\lambda] \geq 0$:

- The pair $(A, B)$ is *stabilizable*: $(A - \lambda I \quad B) = n$ has full row rank.

- The pair $(A, C)$ is *detectable*: $\begin{pmatrix} C \\ A - \lambda I \end{pmatrix} = n$ has full column rank.

The linear system $K$ *strongly stabilizes* $P$ if (i) $K$ stabilizes $P$ and (ii) $E$ is a stability matrix. Necessary and sufficient conditions for the existence of a strongly stabilizing $K$ are presented in reference (37).

## A.4 Decentralized stabilization

Let $P \sim \left( \begin{array}{c|c} A & B \\ \hline C & 0 \end{array} \right)$, with $A$ having dimensions $n \times n$, have the structure

$$B = (B_1 \quad \dots \quad B_k) \ \& \ C = \begin{pmatrix} C_1 \\ \vdots \\ C_k \end{pmatrix}$$

for some integer $k$. Suppose there exist linear systems

$$K_1 \sim \left( \begin{array}{c|c} E_1 & F_1 \\ \hline G_1 & H_1 \end{array} \right), \dots, K_k \sim \left( \begin{array}{c|c} E_k & F_k \\ \hline G_k & H_k \end{array} \right)$$

such that the combined linear dynamics

$$\dot{x} = Ax + (B_1 \quad \dots \quad B_k) \begin{pmatrix} u_1 \\ \vdots \\ u_k \end{pmatrix}$$

$$\dot{\xi}_1 = E_1\xi_1 + F_1 y_1$$
$$\vdots$$
$$\dot{\xi}_k = E_k\xi_k + F_k y_k$$
$$u_1 = G_1\xi_1 + H_1 y_1$$
$$\vdots$$
$$u_k = G_k\xi_k + H_k y_k$$
$$y_1 = C_1 x$$
$$\vdots$$
$$y_k = C_k x$$

are stable. Then the $(K_1, ..., K_k)$ achieve *decentralized stabilization* of $P$. The previous conditions of $(A, B)$ stabilizable and $(A, C)$ detectable are necessary, but not sufficient, conditions for decentralized stabilization.

The following theorem from reference (29) provides necessary and sufficient conditions for decentralized stabilization. First, for any partition $Q \cup R = \{1, 2, ..., k\}$ define $B|^Q$ as the matrix formed by extracting the block columns of $B = (B_1 \quad ... \quad B_k)$ with indices in $Q$, i.e.,

$$B|^Q = \begin{pmatrix} B_{q_1} & ... & B_{q_{|Q|}} \end{pmatrix}$$

with $\{q_1, ..., q_{|Q|}\} = Q$. Likewise, define $C|_R$ as the matrix formed from the block rows of

$C = \begin{pmatrix} C_1 \\ \vdots \\ C_k \end{pmatrix}$, i.e.,

$$C|_R = \begin{pmatrix} C_{r_1} \\ \vdots \\ C_{r_{|R|}} \end{pmatrix}$$

with $\{r_1, ..., r_{|R|}\} = R$.

**Theorem A.1** ((29), Theorem 3). *There exist $(K_1, ..., K_k)$ that achieve decentralized stabilization of $P$ if and only if*

$$\mathbf{rank} \begin{pmatrix} A - \lambda I & B|^Q \\ C|_R & 0 \end{pmatrix} = n$$

*for all complex $\lambda$ with $\mathbf{Re}[\lambda] \geq 0$ and all partitions, $Q \cup R = \{1, 2, ..., k\}$.*

The above rank condition must hold for *all* partitions $Q \cup R = \{1, ..., k\}$. If $R = \emptyset$, one recovers the rank condition for (centralized) stabilizability. Likewise, $Q = \emptyset$ results in the rank condition for detectability.

# B  Proofs

## B.1  Proof of Theorem 4.1

We will examine the conditions of Theorem A.1 on the system (13)–(14). We need to inspect the rank of

$$\begin{pmatrix} \mathcal{A} - \lambda I & \mathcal{B}|^Q \\ \mathcal{C}|_R & 0 \end{pmatrix}$$

for all partitions $Q \cup R = \{1, 2, ..., n\}$. Note that the partitions of either $Q = \emptyset$ or $R = \emptyset$ are already covered by Proposition 4.1.

First, note that

$$\begin{pmatrix} \mathcal{A} - \lambda I & \mathcal{B}|^Q \\ \mathcal{C}|_R & 0 \end{pmatrix} = \begin{pmatrix} \mathcal{M} - \lambda I & 0 & \begin{pmatrix} \mathcal{E}_{q_1} & ... & \mathcal{E}_{q_{|Q|}} \end{pmatrix} \\ \mathcal{M} & -(\lambda + 1)I & 0 \\ \mathcal{M}|_R & -I|_R & 0 \end{pmatrix}.$$

Let $\mathcal{M}$ be an $\ell \times \ell$ matrix. Then

$$\ell = \sum_{i=1}^{n} (k_i - 1).$$

We need the rank of the above matrix to be $2\ell$ for all $\lambda$ with $\mathbf{Re}[\lambda] \geq 0$. Because of the presence of $\mathcal{A} - \lambda I$, loss of rank below $2\ell$ is only possible at eigenvalues of $\mathcal{A}$. Since we are only concerned with $\mathbf{Re}[\lambda] \geq 0$, we focus on eigenvalues of $\mathcal{M}$ (which excludes $\lambda = 0$ by hypothesis, since $\mathcal{M}$ is non-singular).

Without affecting the rank, we can multiply the bottom block row by $-(\lambda + 1)$ and add the middle block rows corresponding $R$ to the rescaled bottom block row to get

$$\mathbf{rank} \begin{pmatrix} \mathcal{A} - \lambda I & \mathcal{B}|^Q \\ \mathcal{C}|_R & 0 \end{pmatrix} = \mathbf{rank} \begin{pmatrix} \mathcal{M} - \lambda I & 0 & \begin{pmatrix} \mathcal{E}_{q_1} & ... & \mathcal{E}_{q_{|Q|}} \end{pmatrix} \\ \mathcal{M} & -(\lambda + 1)I & 0 \\ -\lambda \mathcal{M}|_R & 0 & 0 \end{pmatrix}.$$

Switching the top and bottom block rows results in

$$\begin{pmatrix} -\lambda\mathcal{M}|_R & 0 & 0 \\ \mathcal{M} & -(\lambda+1)I & 0 \\ \mathcal{M}-\lambda I & 0 & \begin{pmatrix} \mathcal{E}_{q_1} & \cdots & \mathcal{E}_{q_{|Q|}} \end{pmatrix} \end{pmatrix}.$$

We can now exploit the block triangular structure. The bottom block row provides a row rank of

$$\sum_{q\in Q}(k_q-1),$$

The middle block row provides a row rank of $\ell$. Finally, the top block row provides a row rank of

$$\sum_{r\in R}(k_r-1).$$

The last assertion is because $\mathcal{M}$ is non-singular, by hypothesis, and therefore it has linearly independent rows. Since $Q\cup R=\{1,...n\}$, we have the desired row rank of $2\ell$.

## B.2  Proof of Lemma 5.1

Define

$$H(s)=C(sI-A)^{-1}B.$$

Since $H(s)$ is a rational function, we can write it as

$$H(s)=\frac{p(s)}{q(s)},$$

for polynomials $p$ and $q$ that have no common roots. The assumption that $CA^mB\neq0$ for some $m$ assures that $H(s)$ is not identically equal to zero.

Suppose that for some $\mu$ and $s'$ that is not an eigenvalue of $A$,

$$q(s')+\mu p(s')=0.$$

Then $s'$ is an eigenvalue of $A-\mu BC$, since

$$\begin{aligned} \mathbf{det}\,[s'I-(A-\mu BC)] &= \mathbf{det}\,[s'I-A]\,\mathbf{det}\,[I+\mu(s'I-A)^{-1}BC] \\ &= \mathbf{det}\,[s'I-A]\,(1+\mu C(s'I-A)^{-1}B) \\ &= \mathbf{det}\,[s'I-A]\,(q(s')+\mu p(s'))\frac{1}{q(s')}. \end{aligned}$$

Note that the roots of $q(s)$ are a subset of the roots of $\mathbf{det}\,[sI-A]$. For sufficiently large $|s|$, we can rewrite $H(s)$ as

$$\begin{aligned} H(s) &= \frac{1}{s}C(I-\frac{1}{s}A)^{-1}B \\ &= \frac{1}{s}C\Big(\sum_{k=0}^{\infty}\frac{1}{s^k}A^k\Big)B. \end{aligned}$$

By assumption, $CB=0$ and $CAB=0$, which implies that the first two terms of the series equal zero. Accordingly,

$$\limsup_{|s|\to\infty}|s|^3|H(s)|<\infty.$$

The main implication here is that the degree of $q(s)$ is at least 3 more than the degree of $p(s)$. Root-locus arguments in references (30) and (31) (asymptote rule) imply that

$$q(s)+\mu p(s)$$

has roots with positive real parts for large $\mu$.

## B.3  Proof of Proposition 5.1

The local dynamics matrix in the game $\Gamma(\mu)$ can be written in the form $A - \mu BC$ with

$$
A = \begin{pmatrix}
0 & 0 & 0 & G_1 & -H_1 & 0 & 0 & 0 & 0 \\
-(1+H_3) & 0 & 0 & 0 & 0 & G_3 & -H_3 & 0 & 0 \\
0 & -(1+H_2) & 0 & 0 & 0 & 0 & 0 & G_2 & -H_2 \\
0 & 0 & 0 & E_1 & -F_1 & 0 & 0 & 0 & 0 \\
0 & 0 & 0 & 0 & -1 & 0 & 0 & 0 & 0 \\
-F_3 & 0 & 0 & 0 & 0 & E_3 & -F_3 & 0 & 0 \\
-1 & 0 & 0 & 0 & 0 & 0 & -1 & 0 & 0 \\
0 & -F_2 & 0 & 0 & 0 & 0 & 0 & E_2 & -F_2 \\
0 & -1 & 0 & 0 & 0 & 0 & 0 & 0 & -1
\end{pmatrix}
\quad
B = \begin{pmatrix}
H_1+1 \\ 0 \\ 0 \\ F_1 \\ 1 \\ 0 \\ 0 \\ 0 \\ 0
\end{pmatrix}
\tag{17a}
$$

$$
C = \begin{pmatrix} 0 & 0 & 1 & 0 & 0 & 0 & 0 & 0 & 0 \end{pmatrix}, \tag{17b}
$$

where we reordered the variables according to $(w_1, w_3, w_2, \xi_1, v_1, \xi_3, v_3, \xi_2, v_2)$ for convenience. Following the same arguments in proving Lemma 5.1, if $CA^m B = 0$ for all $m$, then eigenvalues of $A - BC$ are the same as the eigenvalues of $A$. Writing $A$ in block matrix form yields

$$
A = \begin{pmatrix} A_{11} & A_{12} \\ A_{21} & A_{22} \end{pmatrix},
$$

where $A_{11}$ is $3 \times 3$. Notice that $A_{11}$ is strictly lower triangular, and $A_{21}$ is strictly block lower triangular. Now examine

$$
\det\left[ sI - A \right] = \det\left[ sI - A_{11} \right] \det\left[ (sI - A_{22}) - A_{21}(sI - A_{11})^{-1}A_{12} \right].
$$

One can show that $A_{21}(sI - A_{11})^{-1}A_{12}$ is strictly block lower triangular. Therefore, we have $\det\left[ sI - A \right] = \det\left[ sI - A_{11} \right] \det\left[ sI - A_{22} \right]$. Thus, $A$ has eigenvalues at 0 with multiplicity 3 or more because of $A_{11}$. By exponential stability, there exists $m \geq 2$ such that $CA^m B \neq 0$. Since $CB = 0$, and $CAB = 0$, we can now apply Lemma 5.1.

## B.4  Proof of Proposition 6.1

Reference (37) presents a necessary and sufficient condition for strong stabilizability. First, we compute

$$
T(s) = \mathcal{C}(sI - \mathcal{A})^{-1}\mathcal{B} = \frac{s}{s+1}\mathcal{M}(sI - \mathcal{M})^{-1}.
$$

There is blocking zero (i.e., $T(s) = 0$) at $s = 0$ and $|s| \to \infty$. According to reference (37), a necessary and sufficient condition for strong stabilizability is that there should be an *even* number of eigenvalues in between such pairs of real zeros. This property is known as the "parity interlacing principle". The eigenvalues of $\mathcal{A}$ are $-1, -1, \pm\sqrt{m_{12}m_{21}}$, and so there is a single (and hence, an odd number) of real eigenvalue in between two real zeros of $T(s)$.

## C  Destabilization with a sufficiently small $\mu$

The proof in Section 5 can be modified to consider non-convergence over bounded games, e.g.,

$$
\|M_{ij}\|_\alpha < 1 \quad \forall i, j.
$$

Consider first the following lemma.

**Lemma C.1.** *Let $A \in \mathbb{R}^{n \times n}$, $B \in \mathbb{R}^{n \times 1}$ and $C \in \mathbb{R}^{1 \times n}$. Assume that $A$ has eigenvalues at 0 with multiplicity 3 or more. Then for sufficiently small $\mu > 0$, $A - \mu BC$ is not a stability matrix.*

*Proof.* As in the proof of Lemma 5.1, we have

$$
\det\left[ sI - (A - \mu BC) \right] = \det\left[ sI - A \right] (q(s) + \mu p(s)) \frac{1}{q(s)}.
$$

Recall that the roots of $q(s)$ are a subset of the roots of $\det\left[ sI - A \right]$. If $q(s)$ does not have at least 3 roots at zero, then $A - \mu BC$ is not a stability matrix. Otherwise, root-locus arguments in references (30), and (31) (angle of departure rule) imply there exist roots of $q(s) + \mu p(s)$ with positive real parts for small $\mu$. ∎

Using the structure of the local dynamics and the fact that $A$ in (17a) has eigenvalues at 0 with multiplicity 3 or more, one can directly use Lemma C.1 to show the existence of a sufficiently small offending $\mu$.

