# OpenReview forum: "Higher-Order Uncoupled Dynamics Do Not Lead to Nash Equilibrium - Except When They Do"
_NeurIPS.cc/2023/Conference — NeurIPS 2023 poster_

### Official Review · Reviewer_tNmK · 2023-06-19

**Soundness:** 3 good
**Presentation:** 2 fair
**Contribution:** 3 good
**Rating:** 5
**Confidence:** 2

**Summary:**

The paper studies higher order dynamics, i.e., dynamics that can rely on more auxiliary states than those limited by the dimensionality of the action spaces, in network games with pairwise interactions between players. Importantly, these dynamics only depend on the sequence of payoff signals that each player receives and are, thus, called uncoupled (an important property in multi-agent settings). The paper shows that linear versions of such dynamics can locally lead to any isolated mixed Nash equilibrium. However, for each such dynamic, there exists a "simple" anticoordination game for which this dynamic will not work, i.e., will not stabilize the unique isolated mixed NE. Furthermore, as the paper explains, a linear dynamic that converges to a mixed NE may not be meaningful after all.

**Post-rebuttal**: After reading the other reviews and the authors' responses, I conclude that my concerns stem from my limited understanding of the techniques that are used in the paper. In response, I increase my score from 3 to 5 (and the contribution subscore from 2 to 3) to reflect my evaluation of the results, but I decrease my confidence from 3 to 2 to reflect that I didn't not understand (parts of) the techniques used in the paper.

**Strengths:**

- The paper provides existence results for uncoupled dynamics that (locally) converge to mixed Nash equilibria. To do so, the paper creates higher-order dynamics that overcome known limitations of lower-order dynamics, i.e., of dynamics whose dimension is constrained by the dimension of the action spaces.
- The main takeaways of the paper are clearly presented.


**Weaknesses:**

- The paper is hihgly not self-contained. There is a strong reliance to prior literature and to the appendix.
- As a result of the above and of the complicated notation, the paper was very hard for me to read. Although, I couldn't follow some parts and I couldn't verify the derivation of some results, the results seems plausible and the main takeaways are still clear (as mentioned above).
- I found the motivation of main dynamics in line 140 (paragraph 3.2) inadequate - but this may be related to the fact, that the exposition was not good enough for me to be able to follow.
- While the results inform the discussion on paper [16], they are merely existential and quite general. So, their practical and theoretical scope may be rather limited.
- The paper could have done a better job in building upon more recent literature regarding convergence of dynamics to NE in games.

**Questions:**

- Can the authors address the weaknesses mentioned above?
- Line 22: not the most updated list of papers. Here is an indicative reference to help the authors locate more recent papers in the area in my opinion: https://papers.nips.cc/paper_files/paper/2020/file/0ed9422357395a0d4879191c66f4faa2-Paper.pdf
- Line 77: P_i(x_-i) and P(x_-i): I don't understand the subscript i in P_i. Also, the next sentence in lines 77-78 is even more confusing.
- Line 80: is "a" tuple (and other such minor typos - the paper needs a proofreading)
- Equations (4), (5) and (7): I had a hard time to follow the derivation of these equations. This is one instance of my comment above that the paper is highly not self-contained.
- Line 160: wasn't y_i defined above as \dot v_i? Is that the same?
- Line 171: stabilizability and detectability seem to be important but are only defined in the Appendix. I think that the paper needs to be rewritten in a way that it is self-contained and easier to follow.
- Line 241: locally exponentially stable - the same here. This notion has not been defined before.
- Line 288-289: I missed this argument. Is it that the dynamics fail to monotonically improve with respect to input payoffs (line 282)? Again, I couldn't understand the argument without relying on the Appendix.
- Lines 297/304: anticipatory higher-oder learning/passivity, contractive games - another series of terms that are used without having being defined before. At least the conclusions should be accessible by a wider audience, but this is not the case.

**Limitations:**

Yes, adequately.

---

> ### Author Rebuttal · Authors · 2023-08-08
>
> Question>> Can the authors address the weaknesses mentioned above?
>
> See below for an item-by-item discussion.
>
> Question>> Line 22: not the most updated list of papers.
>
> The list of papers, while admittedly brief, is representative and covers the qualitative aspects of convergence or non-convergence. We are glad to visit the suggested reference and papers cited therein. The suggested reference is a nice complement to the present paper in that it contrasts what can be concluded with and without higher-order dynamics.
>
> Question>>  Line 77: I don't understand the subscript in P_i. Also, the next sentence in lines 77-78 is even more confusing.
>
> The subscript $i$ indicates that $P_i$ is the payoff vector of player $i$ which as mentioned has the same dimension as the strategy vector $x_i$. The subscript was mistakenly dropped in line 77. The sentence in line 78 suggests viewing each entry of $P_i$ as a payoff associated to a certain strategy of player $i$.
>
> Question>> Line 80: is "a" tuple (and other such minor typos-
>
> A final version will take care of all the minor editorial issues
>
> Question>> Equations (4), (5), and (7): I had a hard time to follow the derivation of these equations.
>
> As strategies evolve on the simplex, it is beneficial to project dynamics to their natural dimension (e.g., the simplex in R^3 is two-dimensional). The variable $w_i$ represents movement on this low-dimensional subspace. A revision can reinforce this interpretation.
>
> Question>> Line 160: wasn't y_i defined above as \dot v_i? Is that the same?
>
> The specific linear dynamics that ensure compliance with Assumption 2.1 have an internal state denoted by $v_i$ and the equation $\dot{v_i} =  ...... $ describes the evolution of the state $v_i$. The variable $y_i$ is the output of this specific linear system.
>
> Question>> Line 171: stabilizability and detectability seem to be important but are only defined in the Appendix.
>
> Without going into the definitions, an implication of stabilizability and detectability is the existence of *coupled* learning dynamics that lead to Nash Equilibrium. This is a clear prerequisite to the existence of uncouple learning dynamics. A revision can reinforce this point.
>
> Regarding the paper being self-contained, the paper uses both basic concepts from control theory as well as less widely utilized ones (decentralized stabilization and strong stabilization). Accordingly, we believe that a peripheral value of the paper is bringing these concepts to the forefront in multi-agent learning.
>
> Question>> Line 241: locally exponentially stable
>
> The notion is a basic one in the study of dynamical systems. A revision can add a reference.
>
> Question>> Line 288-289: Is it that the dynamics fail to monotonically improve with respect to input payoffs (line 282)?
>
> The issue is that the dynamics fail to converge (monotonically or not) to a best response to *constant* payoff vector. A revision can better explain this point.
>
> Question>> Lines 297/304:  At least the conclusions should be accessible by a wider audience, but this is not the case.
>
> There are two parts to this paragraph. The first is that there is a discussion to be had on what constitutes “natural”. This is stated plainly. The second part, which is less accessible, points to candidate notions with references, some more familiar than others (e.g., no regret vs passivity). A revision can try to broaden the accessibility
>
>
> Weakness>> The paper is highly not self-contained. There is a strong reliance to prior literature and to the appendix.
>
> We are doing our best work within the page limitations to bring into the paper what is essential to the main contributions and refer to prior literature and appendix for background and support. The current approach maintains the flow of the paper while providing a thorough overview in the appendix to help the reader navigate the paper.
>
> Weakness>> As a result of the above and of the complicated notation, the paper was very hard for me to read. Although, I couldn't follow some parts and I couldn't verify the derivation of some results, the results seems plausible and the main takeaways are still clear
>
> We are encouraged to hear that the main takeaways are clear. As mentioned previously, a peripheral value of the paper is that it brings in new tools to multi-agent learning, and we trying to work with the page restrictions accordingly.
>
> Weakness>> I found the motivation of main dynamics in line 140 inadequate
>
> The main dynamics are a special case of the general form in Section 2.3 where (i) the baseline learning rule is gradient-play and (ii) the higher-order modification is restricted to be linear. The payoff vector is processed through specific linear dynamics (call it a preprocessing phase) to ensure compliance with assumption 2.1, and then the output $y_i$ of the preprocessing procedure is used in the general decision dynamics (which are also linear). We can reinforce this interpretation in a revision.
>
> Weakness>> While the results inform the discussion on paper [16], they are merely existential and quite general. So, their practical and theoretical scope may be rather limited.
>
> There is a clear theoretical contribution to delineate what is or is not possible under uncoupled learning rules. Furthermore, the control-theoretic tools used herein enable the analysis of higher-order multi-agent learning beyond widely studied cases such as zero-sum games or pure strategy equilibria. Regarding the statement that the results are merely existential, there are approaches in the control theory literature to construct higher-order dynamics. We chose to omit these constructions since the focus here is on fundamental limits under uncoupled learning.
>
> Weakness>> The paper could have done a better job in building upon more recent literature
>
> We can include more recent literature, such as the paper mentioned by the reviewer. However, none of the recent literature addresses the specific questions of the present paper.

---

> > ### Comment · Reviewer_tNmK · 2023-08-15
> > **Post Rebuttal Acknowledgement**
> >
> > I thank the authors for responding to my comments. After reading their response and the other reviews, I conclude that all of my concerns (all 5 points in the weaknesses that I mentioned and many of the points raised in the questions) stem from my poor understanding of the techniques that are used in the paper. I still think that the authors could have provided a better exposition to aid readers that are familiar with game-theoretic learning dynamics but not necessarily with the tools used in the paper, but I don't insist that the authors should do any particular changes regarding what to include in the main part nor in their references. I only encourage the authors to implement the changes proposed in their response to my comments above that they think will improve their paper. Based on the above, I increase my score from 3 to 5 but I reduce my confidence from 3 to 2.

---

### Official Review · Reviewer_ZZgJ · 2023-06-25

**Soundness:** 3 good
**Presentation:** 2 fair
**Contribution:** 4 excellent
**Rating:** 7
**Confidence:** 2

**Summary:**

This paper studies multi-agent learning dynamics and the central question is if there is an iterative learning process in a multiplayer game that leads to a Nash equilibrium. There has been substantial prior work on this question for many specific games and learning strategies—generally it is not the case that known dynamics lead to a Nash equilibrium and this is perhaps frustrating as it would be convenient to be able to find Nash equilibria this way. The authors approach this problem in a way that is novel, as far as I know, by specifying a class of interesting/acceptable learning dynamics and using tools from feedback control systems to prove properties of that class. For interesting/acceptable, they require that dynamics are "payoff based," which essentially limits the ability of each agent to see "into" the action space of other agents, and, I believe for tractability of the analysis, they limit the dynamics to a kind of generalized higher-order gradient play. They specifically show that
1. If a game has a strictly mixed Nash equilibrium, there exist payoff-based dynamics that converge locally to that NE. (As a consequence, they show that these dynamics also converge to NE of "nearby" games.)
2. However, there is no "overall good" dynamics—for any such dynamics, there exists a game with unique mixed NE such that these dynamics are unstable at that NE.

**Strengths:**

The results are likely to be novel and provide important context to those who study multi-agent learning dynamics. As is typical, there are some asterisks (the particular learning dynamics are general, but I think the broader questions could motivate looking at an even more general class). The negative result about games with a unique NE seems particularly strong.

The authors have done substantial work to make the results interpretable, at least on the surface, to a broader audience that is not familiar with the control-theoretic tools they are using. I would challenge the authors to go even further with this (see weaknesses below).



**Weaknesses:**

For someone who is not familiar with the specific tools the authors use, the paper is quite hard to follow, even if the game-theoretic aspects are clear and familiar. Appendix D has some worked examples with plots in them. I would suggest thinking about whether it is possible to move some of this content to the main paper. The figures in the paper itself are not that helpful and could be improved, potentially with more detail. The paper lacks good figures currently.

Anything the authors can do to further broaden the context of their results would be helpful.

**Questions:**

I would be interested in the authors' thoughts on whether they view the class of higher-order dynamics they study as restrictive or not. It would be helpful to know, of the papers they cite, in which cases their dynamics class subsumes that of the methods proposed by that paper.

**Limitations:**

Yes, this is a theoretical work and the assumptions of the theorems are clearly stated.

---

> ### Author Rebuttal · Authors · 2023-08-08
>
> Question>> I would be interested in the authors' thoughts on whether they view the class of higher-order dynamics they study as restrictive or not. It would be helpful to know, of the papers they cite, in which cases their dynamics class subsumes that of the methods proposed by that paper.
>
> For the convergence results of Section 4, more restrictions strengthen the results in that additional structures are not needed. The framework set forth in Section 2.3 does capture prior classes of higher-order learning dynamics characterized by differential equations. The paper goes on to restrict its attention to higher-order gradient learning.
>
> Weakness>> For someone who is not familiar with the specific tools the authors use, the paper is quite hard to follow, even if the game-theoretic aspects are clear and familiar. Appendix D has some worked examples with plots in them. I would suggest thinking about whether it is possible to move some of this content to the main paper. The figures in the paper itself are not that helpful and could be improved, potentially with more detail. The paper lacks good figures currently. Anything the authors can do to further broaden the context of their results would be helpful.
>
> Working within the page limitations, one possibility is to move the proof sketches to the appendix and bring a couple of examples to the main body. The message of Figure 1 is an important one to make the connection to feedback systems. We will improve the caption to reinforce this message.

---

> > ### Comment · Reviewer_ZZgJ · 2023-08-16
> > **Reponse to the authors**
> >
> > I'm not fully satisfied with the authors' responses (or the paper in general—it is just really difficult). But I will keep my score—I think the paper still has substantial strengths.

---

### Official Review · Reviewer_mLZz · 2023-07-03

**Soundness:** 3 good
**Presentation:** 3 good
**Contribution:** 2 fair
**Rating:** 6
**Confidence:** 2

**Summary:**

This paper studies higher order payoff based learning, and in particular higher order gradient play in this setting. The authors show that for games with isolated completely mixed NE, there are higher-order gradient play dynamics that converge to that NE. Moreover, that same NE is converged to in ‘nearby’ games as well. However, the authors also show that there exist anti-coordination games where higher-order gradient play dynamics fail to converge to NE. Finally the authors also argue that dynamics that do lead to NE in a coordination game must be inherently unstable.

**Strengths:**

I find the paper to be well structured and readable. The results in the context of uncoupled learning dynamics in games are also interesting and meaningfully extend existing ideas into higher order dynamics.

**Weaknesses:**

A weak point in the paper for me is that the results are based upon the assumption that the mixed NE is a practically useful solution concept in learning in games. However, recent works have shown that the NE can often not only be a poor metric for players’ performance, but also it is unnatural for players using decentralized dynamics to converge to an NE in general games. Thus, for a paper that focuses on higher order learning it would have been much more compelling to focus on a more complete picture of higher order gradient dynamics. What characterizes stable equilibria/fixed points for these dynamics in this setting? In cases where NE are not stable, what do the dynamics look like? In my opinion, a broader view of the dynamical system properties would make the results more interesting and useful.

**Questions:**

For the higher order dynamics, is there intuition about the bandit setting where players only observe (potentially random) realizations of their payoffs? This seems more reasonable for the cases where payoff vectors are large/there are a large number of players and complexity is a concern.

**Limitations:**

The authors adequately addressed the limitations of the dynamics studied.

---

> ### Author Rebuttal · Authors · 2023-08-08
>
> Question>> For the higher order dynamics, is there intuition about the bandit setting where players only observe (potentially random) realizations of their payoffs? This seems more reasonable for the cases where payoff vectors are large/there are a large number of players and complexity is a concern.
>
> As noted by the reviewer, a main issue is where payoff vectors are large (a large number of players need not imply a large payoff vector). We believe that the present results can be used to analyze instantaneous scalar payoffs in the case of discrete-time learning with randomized action selection.
>
> The continuous time ordinary differential equations (ODEs) presented in this paper can be seen as the ODEs that emerge from the ODE method of stochastic approximation (e.g., Benaim, “A dynamical system approach to stochastic approximations, 1996) to analyze discrete time stochastic iterations. Prior work (Fudenberg and Levine, “Consistency and cautious fictitious play”, 1995) illustrates how the scalar payoff case of fictitious play can be analyzed using such methods. This approach also was utilized to analyzed higher-order learning under scalar payoffs for the specific case of “anticipatory” higher order learning (Arslan and Shamma, “Distributed convergence to Nash equilibria with local utility measurements”). Likewise, we believe that the case of instantaneous scalar payoffs can be addressed using the setting in the present paper as the basis of the emergent ODEs of stochastic approximation.
>
>
> Weakness>> A weak point in the paper for me is that the results are based upon the assumption that the mixed NE is a practically useful solution concept in learning in games. However, recent works have shown that the NE can often not only be a poor metric for players’ performance, but also it is unnatural for players using decentralized dynamics to converge to an NE in general games. Thus, for a paper that focuses on higher order learning it would have been much more compelling to focus on a more complete picture of higher order gradient dynamics. What characterizes stable equilibria/fixed points for these dynamics in this setting? In cases where NE are not stable, what do the dynamics look like? In my opinion, a broader view of the dynamical system properties would make the results more interesting and useful.
>
> The question of the relevance of Nash Equilibrium is a long standing one that won’t be resolved in this paper. Nonetheless, interest in Nash Equilibrium remains widespread. Also, studying higher-order uncoupled dynamics is an interesting topic on its own. We further believe that these results may be relevant to settings other than mixed-strategy Nash equilibrium, such as higher-order multi-agent learning in the absence of strict convexity.
>
> As recognized by the reviewer, recent work shows the lack of natural decentralized dynamics to converge to a NE in general. This paper contributes to that line of work in relaxing the requirement of generality (Section 4), but also showing the lack of universality (Section 5). Furthermore, the topic of strong stabilization reinforces that the lack of natural dynamics to converge to the mixed equilibrium of a coordination game (Section 6).
>
> Regarding the characterization of stable equilibria (beyond eigenvalues), the results of this paper establish that it depends on both the specifics of the game and the structure of higher-order dynamics.

---

> > ### Comment · Reviewer_mLZz · 2023-08-16
> > **Response to Author Rebuttal**
> >
> > Thank you for your clarifications and explanations. It clears up some doubts I had about the paper and if the authors would add some further explanation and comparison of their dynamics with previous work and clarify their contributions in the paper, I think the paper would be very suitable for NeurIPS.
> >
> > Best regards,
> >
> > Reviewer mLZz

---

### Official Review · Reviewer_VfQG · 2023-07-06

**Soundness:** 3 good
**Presentation:** 3 good
**Contribution:** 3 good
**Rating:** 6
**Confidence:** 4

**Summary:**


The paper shows that for any finite game with an isolated completely mixed Nash Equilibrium, there exist a payoff based higher-order
gradient play dynamics that lead (locally) to that Nash equilibrium, both for this game and all payoff nearby games. Conversely, they show that for
any higher-order gradient play dynamics, there is a game with a unique isolated completely mixed Nash equilibrium for which that dynamics do not locally converge to that Nash Equilibrium.

 Hart and A. Mas-Colell proved using an anti-coordination game that no first-order uncoupled dynamics leads to the unique interior Nash equilibrium of that game.  Shamma and Arslan (2005) proved that there are higher order dynamics which leads to that equilibrium. This paper shows that this extends to all games.


**Strengths:**

Tools used are, up to my knowledge, not standard in evolutionary game theory: decentralized stabilizing control and root-locus which characterizes the locations of the eigenvalues of a matrix as a function of a scalar parameter!

**Weaknesses:**


The biggest weakness for me is that there is no micro-foundation of the class higher order dynamics nor a comparison with the previously studied higher order dynamics.


**Questions:**


 - Are your dynamics more general than all the previously studied one (such as [18, 19, 20, 23] etc)?
 - Do you have any micro-foundation of your higher order dynamics?


**Limitations:**

The converse result (for any higher-order dynamics, there is a game with a unique isolated completely mixed Nash equilibrium) is proved only for gradient play dynamics and not all higher order dynamics !

---

> ### Author Rebuttal · Authors · 2023-08-08
>
> Question>> Are your dynamics more general that all the previously studied ones (such as [18, 19, 20, 23] etc)?
>
> The above papers are specific instances of higher-order dynamics. Setting aside continuous-time/discrete-time differences, the framework of higher-order learning outlined in Section 2.3 is more general. The convergence results of Section 4 shows that higher-order gradient play is sufficient to lead to any mixed-equilibrium, and so more generality is not needed.
>
>
> Question>> Do you have any micro-foundation of your higher-order dynamics?
>
> The issue of micro-foundations depends on the specifics of the higher-order dynamics. The structure presented in Section 2.3 allows a higher-order augmentation to existing learning rules for which there are micro-foundations (e.g., Sandholm, “Population Games and Deterministic Evolutionary Dynamics”, in Handbook of Game Theory with Economic Applications, 2015). These learning rules have agents reacting to the payoffs, and higher-order dynamics can capture path dependent phenomena in these payoffs such as recency bias (e.g., Fudenberg and Levine, “Recency, consistent learning, and Nash equilibrium”, 2014).
>
> Weakness>> The biggest weakness for me is that there is no micro-foundation of the class higher order dynamics nor a comparison with the previously studied higher order dynamics.
>
> See above response to the questions raised by this reviewer.

---

> > ### Comment · Reviewer_VfQG · 2023-08-11
> >
> > I thank the authors for their reply. However, they don't completely solve my concerns. For example, the authors in [23] spent some effort to justify their dynamics, even if they are a natural extension of the replicator dynamics. It would be of interest if the authors investigate seriously the foundational question. Also, when they claim that all the previous dynamics are a particular case of their dynamics, I can believe it but adding some examples in this direction would improve the paper.
> >
> > That said, I believe this is a good paper, which contains some important contributions and technics and so is a good candidate for Neurips.

---

### Official Review · Reviewer_f6g4 · 2023-07-06

**Soundness:** 4 excellent
**Presentation:** 4 excellent
**Contribution:** 4 excellent
**Rating:** 8
**Confidence:** 3

**Summary:**

This paper shows the lack of universality on the side of both games and learning dynamics (even for higher-order ones)! Particularly, for any game with a mixed-strategy Nash equilibrium (NE), there exists uncoupled payoff-based (possibly high-order) dynamic converging locally to the NE. However, any such dynamics can also be destabilized by a suitable anti-coordination game. Notably, the paper uses classical analysis methods in feedback control systems. Highlighted similarities between higher-order learning in games and higher-order optimization algorithms, such as momentum-based or optimistic gradient algorithms, are also interesting.

**Strengths:**

The widely studied fictitious play dynamics are known to converge equilibrium in many interesting games but not all of them, e.g., see Shapley's counterexample. Therefore, researchers were looking for a learning dynamic that can converge to equilibrium in every game to justify equilibrium analysis. However, Hart and Mas-Colell, Ref. (16), proved the negative result that there does not exist (first-order) uncoupled learning dynamics that can converge to equilibrium in anti-coordination games, and therefore, there cannot be universally convergent (uncoupled) learning dynamics.

Later, Shamma and Arslan, Ref. (17), showed that higher-order learning dynamics can converge to equilibrium in anti-coordination games. This paper provides a more general result saying that for any game (with a mixed-strategy Nash equilibrium), there exists a (possibly high-order) payoff-based learning dynamic that can converge locally to that equilibrium. Seeing this, researchers may start looking for universally convergent higher-order learning dynamic that can converge to equilibrium in every game. However, the paper proves the negative result that given any such higher-order dynamics, there always exists a certain anti-coordination game in which the dynamics do not converge to equilibrium. Therefore, we can view this paper as a generalization of Hart and Mas-Collel's seminal negative result to higher-order learning dynamics.

Note that Foster and Young designed uncoupled stochastic rules, known as regret testing, that can converge probabilistically to Nash equilibrium in every two-player strategic-form game. However, the convergence is in the relatively weak sense by saying that players will be at equilibrium most of the time though they may move away from it.

Note also that complexity results related to Nash equilibrium computation are not relevant since the paper focuses on asymptotic convergence for finite games. A mixed-strategy Nash equilibrium always exists in strategic-form games, with finitely many player and action.

Because of these reasons, I believe that this result is worth being taught in (advanced) game theory courses.

I acknowledge that I have read the rebuttal.

**Weaknesses:**

- Figures might include captions with more detailed descriptions.

**Questions:**

- What is the main obstacle to address instantaneous scalar payoffs rather than payoff vector setup?

**Limitations:**

The limitations are highlighted explicitly.

---

> ### Author Rebuttal · Authors · 2023-08-08
>
> Question>> What is the main obstacle to address instantaneous scalar payoffs rather than payoff vector setup?
>
> We believe that the present results can be used to analyze instantaneous scalar payoffs in the case of discrete-time learning with randomized action selection.
>
> The continuous time ordinary differential equations (ODEs) presented in this paper can be seen as the ODEs that emerge from the ODE method of stochastic approximation (e.g., Benaim, “A dynamical system approach to stochastic approximations, 1996) to analyze discrete time stochastic iterations. Prior work (Fudenberg and Levine, “Consistency and cautious fictitious play”, 1995) illustrates how the scalar payoff case of fictitious play can be analyzed using such methods. This approach also was utilized to analyzed higher-order learning under scalar payoffs for the specific case of “anticipatory” higher order learning (Arslan and Shamma, “Distributed convergence to Nash equilibria with local utility measurements”, 2004). Likewise, we believe that the case of instantaneous scalar payoffs can be addressed using the setting in the present paper as the basis of the emergent ODEs of stochastic approximation.
>
> Weakness>> Figures might include captions with more detailed descriptions.
>
> The final version will revisit the figure captions to add more details.

---

### Decision · Program_Chairs · 2023-09-21

**Decision:**

Accept (poster)

**Comment:**

This paper examines the (im)possibility of convergence to Nash equilibrium in polymatrix games under higher-order learning dynamics. The authors' main result is that, for every polymatrix game with an isolated, fully mixed Nash equilibrium, there exists a higher-order gradient-based dynamical system for which this equilibrium is stable. On the flip side, the authors also show that for a specific, projection-type dynamical system, the mixed Nash equilibrium of Jordan's matching pennies game is unstable.

The reviewers appreciated the paper's technical contributions and most of the reviewers' concerns were addressed satisfactorily during the discussion phase. The main issues that were raised concerned the paper's readability from a wider audience, and the fact that no microfoundations (or other type of justification) were given for the dynamics that "detect" a given mixed equilibrium.

The readability issue is mostly due to the fact that the authors are employing notions and terminology from control theory, which are not sufficiently standard in the field at large; for this, I would urge the authors to take advantage of the extra page to provide more context and explanations for some of the lesser-known tools and techniques that they employ (including the definition of the dynamics, the notion of "washout filters", etc.).

The microfoundations issue is more tricky: since this is an existence result, the authors' approach is justified in juxtaposition to the impossibility result of Hart & Mas-Colell; at the same time, if the dynamics that detect a given mixed equilibrium only work for a particular game and are otherwise non-convergent, then, in the absence of solid microfoundations, the authors' existence result is somewhat weakened.

Regardless, even with the above limitations, the paper's analysis and results provide a number of fruitful insights into a difficult problem, so there were no objections to an "accept" recommendation. My own assessment is aligned with the reviewers' consensus, and I am happy to recommend acceptance as well.

I would only like to draw attention to the following point: the authors' analysis concerns *polymatrix* games, not the mixed extension of finite games. This does not weaken the paper's contributions, but it does mean that the paper's setting is not quite the same as classical finite games. This should be made clearer in the abstract and the introduction as, otherwise, it could lead to confusion in the literature.